# Force-dependent development of the myodural bridge in rats: The impact of Integrin α7

**Lu Zhang[1]◉, Yun-Feng Liu[1,2]◉, Mi Luo[1], Xue Song[1], Xin-Yuan Zhang[1], Jing-Xian Sun[3], Jian-Fei Zhang[1], Xiao-Ying Yuan[1], Yan-Yan Chi[1], Ruo-Tong Zhang[1], Chan Li[1], Campbell Gilmore[4], Sheng-Bo Yu[1], Wei Ma [1]*, Hong-Jin Sui[1]***

**1** Department of Anatomy, College of Basic Medicine, Dalian Medical University, Dalian, China, **2** Department of Anatomy, College of Basic Medicine, Jilin Medical University, Jilin, China, **3** College of Fisheries and Life Science, Dalian Ocean University, Dalian, China, **4** Flinders Medical Centre, Adelaide, Australia

◉ These authors contributed equally to this work.
* mawei_dlmedu@163.com (WM); suihj@hotmail.com (H-JS)

## Abstract

The myodural bridge (MDB) represents specialized fibrous structures establishing connectivity between suboccipital musculature and the spinal dura mater (SDM). The suboccipital muscles, ligaments, and myodural bridge fibers together form a functional unit known as the myodural bridge complex (MDBC). Mechanical stress from suboccipital muscles may contribute to MDB maturation. Integrin α7 (ITGA7) is critical for skeletal muscle attachment to connective tissues, and is involved in the transmission of lateral and longitudinal forces in skeletal muscle. Given the muscle force transmission characteristics of ITGA7 and the dependence of MDB development on force transmission, we hypothesized that ITGA7 serves as a crucial link between RCDmi and the MDB it emits, and may involve in the development of MDBC. To test this, neonatal Sprague-Dawley (SD) rats were randomly allocated to shRNA-ITGA7, shRNA-NC control, lentiviral vectors were injected into the dorsal atlanto-occipital interspace. ITGA7 suppression significantly impaired MDB development and maturation, manifesting as disrupted fiber assembly and RCDmi muscle dystrophy. Ultrastructural analysis revealed disorganized collagen fiber architecture and an abundance of fibroblasts, indicative of immature collagen fibers, further corroborated by Picrosirius red staining. Additionally, ITGA7 knockdown resulted in diminished RCDmi muscle force and altered ECM-related gene expression profiles. A key finding of our study is the importance of ITGA7 as a direct molecular link between suboccipital muscles and MDB, suggesting that mechanical forces from suboccipital musculature fundamentally influence MDB differentiation and maturation. These findings substantiate MDB's role in force transmission to the SDM and by extension, advance our understanding of the molecular mechanisms underlying MDB development and its physiological significance.

**Data availability statement:** All relevant data are within the manuscript and its Supporting information files.

**Funding:** This work was supported by the following funding sources: • National Natural Science Foundation of China (NSFC32471192, NSFC32100928) 。 NSFC32471192: Hongjin Sui 。 NSFC32100928: Xiaoying Yuan 。 Website: https://www.nsfc.gov.cn/ • Science and Technology Talent Innovation Support Policy Project of Dalian (2023RG003) 。 2023RG003: Hongjin Sui 。 Website: https://www.dl-rc.com • Youth Talent Cultivation Fund Project of Dalian Medical University 。 Recipient: Ruotong Zhang 。 Website: https://jwc.dmu.edu.cn. The funders had no role in study design, data collection and analysis, decision to publish, or preparation of the manuscript.

**Competing interests:** The authors declare no competing interests.

## Introduction

Cerebrospinal fluid (CSF) circulation, recognized as the body's third circulatory system, encompasses a complex sequence of processes from choroid plexus production through cerebral ventricles, subarachnoid space, and parenchyma, culminating in venous reabsorption [1]. This dynamics system is influenced by several factors, including arterial pulsations, respiratory movements, body position [2–4], glymphatic circulation, circadian rhythms [5–7], et al. Notably, the upper cervical region harbors a specialized biomechanical microenvironment that may be involved in localized CSF regulation— the myodural bridge (MDB) [8–11].

The MDB consists of fibrous connective tissues that runs between the suboccipital muscles and the cervical spinal dura mater (SDM), passing through the atlanto-occipital and atlanto-axial interspaces [12,13]. The MDB is composed of fibers from multiple sources, including the rectus capitis posterior minor (RCPmi), rectus capitis posterior major (RCPma), obliquus capitis inferior (OCI), nuchal ligament, nuchal-dural bridge (previously referred to as TBNL), and vertebrodural ligament [14–21]. These muscles, ligaments, and the MDB fibers collectively form an integrated functional unit known as the myodural bridge complex (MDBC) [22].

Comparative anatomical studies have demonstrated that the MDB is a highly conserved structure with significant physiological functions in Vertebrate [12,23–26]. Its physiological functions include influencing CSF circulation, preventing dura mater folding, transmitting proprioception, maintaining subarachnoid space integrity, and associating with cervicogenic headache [16,27,28].

The MDB is a tendinous-like structure primarily composed of parallel arranged collagen type I fibers [29]. similar to tendons, its strong collagen fibril arrays enable efficient force transmission from the suboccipital muscles to the SDM [29,30]. Studies in developmental biology have elucidated the MDB development process in rats and human embryos. In rat embryos (E18-E21), only few fibers and muscles were present in the suboccipital region. An obvious increase in the number of fibers and muscle tissues was noted at P7. MDB fibers gradually became denser and more organized at P14. In human fetuses, the muscle cells matured at 19 weeks of gestation. The MDB cells became oriented and cross the atlanto-occipital interspace, attaching to the dura mater at 21 weeks of gestation. It is speculated that mechanical stress from suboccipital muscles may contribute to MDB maturation [31–33]. Consequently, further direct experimental research is essential to confirm this hypothesis. Moreover, research on the molecular mechanisms that regulate the MDB formation is limited.

Integrins have become fundamental cell adhesion receptors, mediating cell and tissue functions in diverse health and disease scenarios [34,35]. They attach the cytoskeleton of cells to specific proteins in the extracellular matrix (ECM), being crucial for cell migration, cell shape determination, and cell-cell interactions. The force transmission pathway, ECM-integrins-adaptor proteins-actin, is well-established. Fluorescence resonance energy transfer measurements confirmed that adaptor proteins such as talin [36,37] and vinculin [38] indeed under force within integrin adhesion complexes. The α7β1 integrin is the predominant integrin on skeletal muscle. It binds

to laminin in ECM, as well as to myofilaments via Talin [39]. In developed muscle fibers, the α7β1 integrin provides both terminal and peripheral cohesion to muscle fibers, which is important to muscle integrity, neuromuscular connectivity, and force generation. In adult muscle fibers, integrin α7 (ITGA7) is predominantly located peripherally and enriched at myotendinous and neuromuscular junctions [40]. ITGA7 plays a crucial role in transmitting lateral forces between skeletal muscle cells and the basement membrane, as well as longitudinal forces at the tendon junction [41]. The absence of ITGA7 results in progressive muscle dystrophy accompanied by increased endomysial connective tissue production [42].

Although the role of the MDB in transmitting suboccipital muscle strength has been confirmed, some research suggested that force transmission via MDB was limited or ineffective [43]. Given the force – related characteristics of ITGA7 and the dependence of MDB development on force transmission, we hypothesized that ITGA7 serves as a crucial link between RCDmi and the MDB it emits. We reasoned that if the MDB transmits lateral forces from the RCDmi during development, then disrupting or weakening this connection would impair MDB formation due to reduced muscle force. Conversely, if the MDB does not function in force transmission from the RCDmi, its formation would remain unaffected. To test this hypothesis, the ITGA7 shRNA lentivirus vehicles were injected into the dorsal atlanto-occipital interspace of neonatal rats to reduce ITGA7 expression, and thereby disrupt its role in force transmission. The impact of this intervention on the morphology of MDB development was examined. This research provides direct evidence supporting the hypothesis that MDB formation is dependent on mechanical loading from suboccipital muscles. Furthermore, this study establishes a foundation for future research aimed at elucidating the molecular regulatory mechanisms and function of the MDBC.

## Materials and methods

### Ethical statements

The experimental protocol was approved by the Committee on the Ethics of Animal Experiments at Dalian Medical University. All experiments were conducted in accordance with the guidelines and regulations of Dalian Medical University. (No: AEE21085). This study was carried out in compliance with the ARRIVE guidelines.

### Animal models and housing

Newborn Sprague Dawley (SD) rats (P0, < 12h old, 6–8 g), derived from adult SD rats (8–12 weeks old, 300–500 g) housed under controlled conditions (20–22°C, 50–70% humidity, 12h light cycle) with ad libitum access to standard rodent diet (supplied by Medicience Ltd., Jiangsu, China) and water. Mixed sex (male/female) were used. All animals exhibited normal activity and weighed 6–8 g. The experimental group and control group were injected with shRNA-ITGA7 lentivirus (targeting Integrin α7) and shRNA-NC lentivirus (non-targeting control), respectively, into the rectus capitis dorsal minor (RCDmi) muscle. A sham group (n = 4) was included to exclude non-specific effects of the injection procedure. P0 rats were randomized by weight and litter using a computer-generated sequence. Each animal received a unique identifier, and group assignments were determined by a predefined random number table. Cage locations were rotated weekly to minimize environmental bias. For euthanasia, rats were exposed to 100% carbon dioxide via a sealed chamber, until complete loss of consciousness was confirmed. The exposure was continued for an additional 3 minutes to ensure death, followed by verification of absent heartbeat and respiration. All possible measures were taken to minimize any potential discomfort or pain experienced by the animals involved.

### Lentivirus packaging and injection

HEK 293T cells were kindly provided by Yiranmingyu Biological Techmology Co, Ltd. Cells were cultured in DMEM (Hyclone, USA) supplemented with 10% fetal bovine serum (GIBCO, USA), 100 U/mL penicillin, and 100 U/mL streptomycin at 37 °C and 5% $CO_2$. Transfection was performed when cell confluence reached 70% to 80%. The shRNA vector

targeting rat ITGA7 (NM_030842) was constructed by Genechem (Shanghai, China), the shRNA sequences were shown in S1 Table. The vector information was hU6-MCS-CBh-gcGFP-IRES-puromycin. The pCDH-CMV-EF1-puro lentivirus packaging system was utilized to generate the GFP-shRNA-ITGA7 lentivirus. The supernatant was collected 36~48 hours after transfection and centrifuged at 4 °C and a speed of 50000g for 30 minutes to obtain high titers of lentivirus. The injected lentivirus was concentrated to titers of $10^9$ IU/mL.

P0 rats were anesthetized with 2% tribromoethanol (0.2 ml/10 g body weight, intraperitoneal injection). Bilateral injections were performed at the the dorsal atlanto-occipital interspace, 1 mm caudal to the occipital protuberance and 1 mm lateral to the midline. A 33-gauge needle was inserted 2 mm deep, and 10 μL of virus was delivered per side using a microinjector (World Precision Instruments). After injection, P0 rats were placed on a 37 °C heating pad for 2 hours then returned to their dams for recovery (Fig 1A). Viral injection personnel were unblinded due to biosafety requirements, but all morphological (e.g., histology) and molecular (e.g., Western blot) analyses were performed by investigators blinded to group assignments. Data were anonymized prior to analysis, and cages were coded to prevent observer bias.

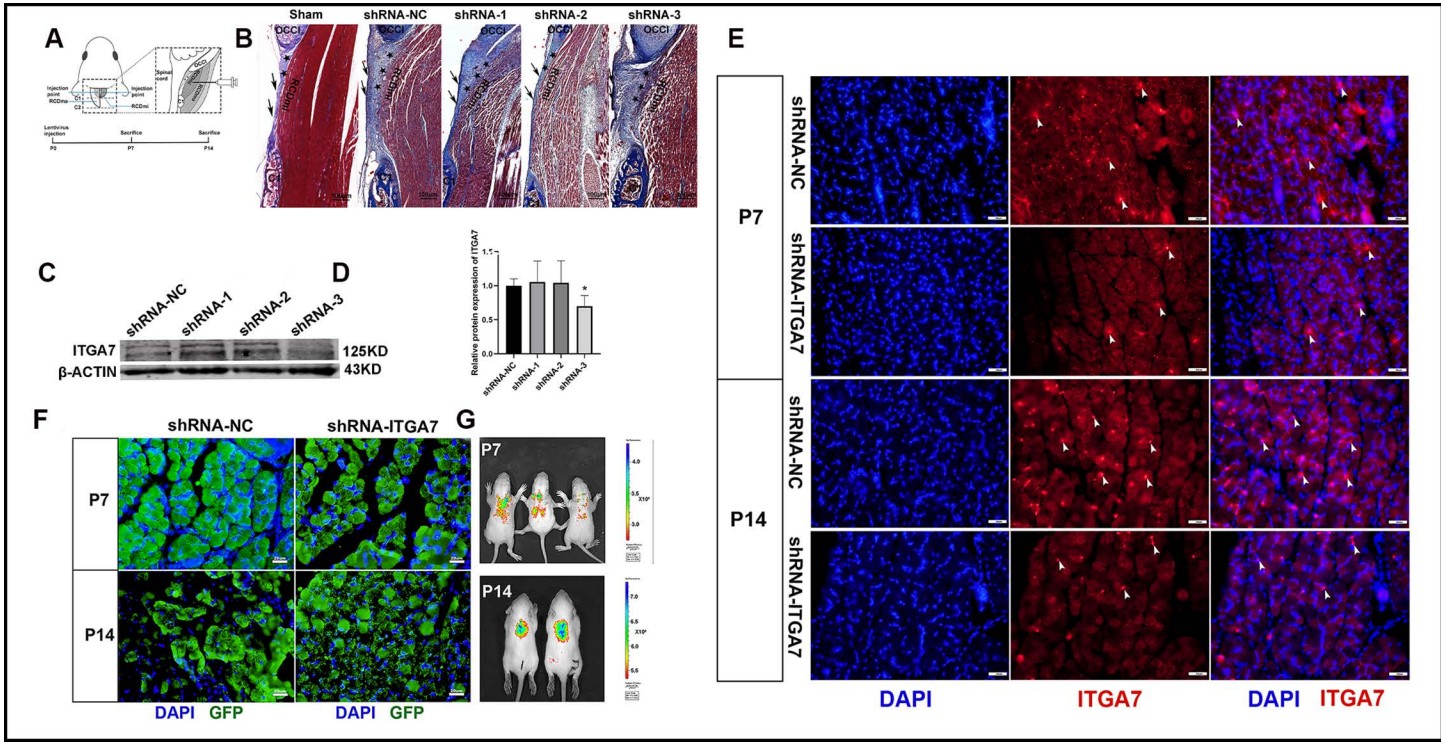

**Fig 1. Localized ITGA7 knockdown in the dorsal atlanto-occipital interspace of Rats.** (A) Schematic diagram of lentivirus injection and sampling time points. (B) Masson staining of the sagittal section of MDBC in the dorsal atlanto-occipital interspace, performed 7 days post- lentivirus injection. The shRNA-NC served as the negative control for the shRNA lentivirus vector, whereas shRNA-1, shRNA-2, and shRNA-3 represent three distinct shRNA lentivirus vectors targeting ITGA7. A Sham group, subjected to identical surgical procedures but without vector injection, was included to exclude potential confounding effects of needle insertion. (C) Western blotting to evaluate ITGA7 expression in MDBC 7 days post-lentivirus injection. (D) Quantitative analysis using Image J, with β-ACTIN as an endogenous control. * $p < 0.05$ (n = 3). (E) Immunofluorescence reveals the expression of ITGA7 in RCDmi within the atlanto-occipital interspace of rats. (F) Immunofluorescence reveals the expression of Green Fluorescent Protein (GFP) in RCDmi within the atlanto-occipital interspace of rats. (G) Representative in vivo images depicting GFP expression of shRNA-ITGA7 vector in rats at P7 and P14. OCCI: occipital bone. C1: atlas. RCDmi: rectus capitis dorsal minor muscle. ★: MDB fibers. ↑: SDM. white arrow: ITGA7 expression.

## Sample size determination

Sample sizes (n = 29/group) were determined based on pilot studies and power analysis using G*Power (α = 0.05, β = 0.8, effect size = 1.5). Initial enrollment included 32 rats per group, with 3 excluded post-GFP screening. P0 rats with successful lentiviral infection confirmed by in vivo GFP fluorescence imaging in the RCDmi region. Undetectable or subthreshold GFP expression, abnormal weight (<6 g or >8 g), are excluded.

## Quantitative real time polymerase chain reaction (qRT-PCR)

Total RNA was extracted using a total RNA isolation kit (Vazyme, China). Total RNA was quantified using a NanoDrop2000 spectrophotometer (Thermo Fisher Scientific, USA). cDNA was synthesized from each RNA sample using HiScript II Q RT SuperMix (Vazyme, China). qRT-PCR was performed using SYBR qPCR Master Mix (Vazyme, China), and the target gene expression levels were quantified relative to the housekeeping gene GAPDH using the $2^{-\Delta\Delta CT}$ method. Tissue sampling sites and methods are detailed in S1 Fig. The primer sequences used are listed in S2 Table.

## Western blotting

Tissues were lysed using total protein extraction kit (Keygen, China). Equal amounts of samples were electrophoresed on SDS-PAGE gel and transferred to nitrocellulose membrane (Invitrogen, CA, USA). The membranes were blocked with 5% (w/v) BSA for 2 hours at room temperature, then incubated with the primary antibody overnight at 4°C, followed by the secondary antibody for 1 hour at room temperature, and detected by Odyssey CLx image system (LI-COR, USA). The primary antibodies used were anti-ITGA7 (#bs-1816R, Bioss, China) at a dilution of 1:1000, and anti-β-Actin (#21338, SAB, USA) at a dilution of 1:5000. The secondary antibody used was Dylight 800 Goat Anti-Rabbit IgG (#A23920, Abbkine, China) at a dilution of 1:2000 dilution.

## In vivo imaging

An in vivo imaging system (IVIS Spectrum, PerkinElmer, USA) was used to examine the GFP expression of shRNA-ITGA7 and shRNA-NC in living rats. Rats at P7 and P14 were first anaesthetized (2% tribromoethanol solution, 0.2 ml/10g body weight) and placed in the shooting box in an adjusted position. This in vivo imaging system can provide us with information about whether shRNA lentiviral vectors are effective and where they are expressed. 3 rats in each group did not show strong GFP expression, and thus were excluded from the analysis.

## Histological sectioning and quantitative analysis

Histological sectioning and staining were conducted on lentivirus-injected rats at P7 and P14, respectively (n = 4). After fixation and decalcification, the specimens were embedded in paraffin wax and sectioned sagittally at 8μm thickness by using a rotary microtome (Leica Micro HM450; Leica Microsystems GmbH, Wetzlar, Germany). Masson's trichrome staining (Masson) and Picrosirius red staining were performed respectively. The sections stained with Masson were photographed using a Nikon NIS image system (Nikon Eclipse 80i, Nikon, Tokyo, Japan). The sections stained with Picrosirius red were observed under a polarizing microscope (Olympus BH-2; Olympus Corp., Tokyo, Japan).

Masson-stained slides from each group were observed and photographed under a light microscope, including MDBC fibers, the dorsal atlanto-occipital membrane (DAOM), SDM, and RCDmi within the dorsal atlanto-occipital interspace. Semi-quantitative analysis was performed by using the Image J software to evaluate the collagen volume fraction as previously described [44]. For Sirius red-stained slides, images were captured under polarized light. Red/orange collagen fibers (type I) were quantified by calculating their area percentage relative to the total tissue area using Image J (RGB threshold settings: R > 180, G < 150, B < 100). The integral optical density (IOD) was recorded to assess staining intensity(n = 4).

## Immunostaining

The RCDmi muscles were quickly frozen, embed them with OCT (BL557A, Biosharp), and made frozen sections. The sections were first rinsed in PBS. They were fixed in 4% paraformaldehyde for 30 minutes, and blocked in 5% serum for 2 hours. The slides were incubated overnight with the primary antibody Integrin a7 (E-2) (1:100, sc-515716, Santa Cruz) and Laminin-α2 (1: 100, sc-59854, Santa Cruz). After rinsing with PBS, the slides were incubated with goat anti-mouse IgG (H+L) rhodamine (TRITC, 1:200, #BS11502, Biogot) for 1 hour. After staining with DAPI, the slides were mounted with mounting medium. The staining were visualized under a fluorescence microscope (DMI4000B, Leica, Germany) (n=3).

## Uniaxial tensile test

After collecting the RCDmi specimens from each group, elastic bandages were applied to the cranial and caudal sides of RCDmi, and then clamped in the fixtures. The uniaxial tensile test was performed using a small soft tissue biomechanical property tester (self-invented device, patent number: 202410169009.9, see S2 Fig) at a tensile speed of 3 mm/min. The maximum breaking forces were collected (n=3).

## Scanning and transmission electron microscopy

The structures of MDBC in the dorsal atlanto-occipital interspace, including MDBC fibers, DAOM, SDM, and RCDmi were fixed in 2.5% glutaraldehyde (Sigma, USA), and post-fixed in 1% OsO4 (Sigma, USA). The specimens for scanning electron microscope (SEM) were dehydrated in a graded series of tert-butanol, then dried at the critical point, coated with platinum, and observed with a scanning electron microscope (JSM-IT300LA, JEOL Ltd., Japan). Specimens for transmission electron microscope (TEM) were dehydrated in a graded series of ethanol and embedded in epoxy resin (Okenshoji, Tokyo, Japan). Ultrathin sections were cut with PT-X Power Tome ultramicrotome (RMC, Arizona, USA) and stained with uranyl acetate and lead citrate. The tissue sections were observed under a transmission electron microscope (JEM-2000EX, JEOL Ltd., Japan) operated at 120 KV(n=3).

## Statistical analysis

Data were analyzed using SPSS25.0 and Image J. Results are presented as mean±standard deviation (SD). Statistical significance between groups was determined using a two-tailed Student's t-test, with $p < 0.05$ considered significant. All experiments were performed at least in triplicate.

## Results

### Screening and validation of an ITGA7 knockdown model of MDBC in rats

We evaluate Itga7 expression in rats at four critical stages: embryonic day 16 (E16), postnatal day 0 (P0), postnatal day 7 (P7), and postnatal day 14 (P14). The expression of Itga7 increased progressively and significantly during MDBC development (* $p < 0.05$, ** $p < 0.01$, S3 Fig), suggesting its critical role in regulating MDBC development.

A rat model with localized ITGA7 knockdown was constructed by injecting shRNA-ITGA7 lentivirus vectors into the MDBC within the dorsal atlanto-occipital interspace. The lentivirus vectors included three shRNAs, labeled as shRNA-1, shRNA-2, shRNA-3, and a negative control, labeled as shRNA-NC. A Sham group (n=4), subjected to identical surgical procedures but without vector injection, was included to exclude potential confounding effects of needle insertion. Samples were taken from animals in each group at P7 to identify the most effective shRNA vector. Compared with the shRNA-NC group, Masson staining exhibited an increase in the spacing and interstitium between RCDmi muscle fibers in all shRNA-ITGA7 groups. In the shRNA-NC group, the MDB fibers ran parallel, extending from the ventral part of the RCDmi to fuse with the DAOM and SDM, interspersed with oriented fibrocytes and fibroblasts. In the shRNA-1 group,

the MDB fibers in the upper part of the atlanto-occipital space exhibited characteristics similar to those in the shRNA-NC group. However, the MDB fibers in the lower part appeared more mature with fewer fibroblasts. In the shRNA-2 group, the MDB fibers were the most mature among all other groups, tightly fused with the DOAM and SDM. In the shRNA-3 group, there was an increase in fiber quantity within the atlanto-occipital space, with fibers and cells displaying a disorganized arrangement (Fig 1B). Total protein was extracted from the MDBC in each group, and Western blotting analysis revealed a consistent downregulation of ITGA7 protein expression levels in the shRNA-3 group (Fig 1C, D). Given the morphological changes observed in the shRNA-3 group, the shRNA-3 vector demonstrated the most efficient knockdown effect. Therefore, shRNA-3 vector was selected in subsequent experiments and then labeled as shRNA-ITGA7.

To evaluate the efficiency of lentivirus infection, the expressions of ITGA7 and shRNA ITGA7 fused GFP were detected in the RCDmi of P7 and P14 rats, respectively. Immunofluorescence showed that the expression of ITGA7 in the RCDmi was lower in the shRNA-ITGA7 group than in the shRNA-NC group at P7 and P14 stages (Fig 1E). In addition, GFP was expressed in the RCDmi in both groups. GFP expression decreased with time (Fig 1F). It is speculated that during the developmental process, the proportion of infected cells will decrease, or some of the infected cells will be eliminated. Meanwhile, an in vivo imaging system was used to examine the GFP expression in shRNA-ITGA7 and shRNA-NC groups of P7 and P14 rats (Fig 1G). The heat map image represented GFP intensity for representative rats. Any rats with weak GFP intensity in the dorsal neck region, such as the third rat at P7, was excluded from the study. GFP intensity observed via in vivo imaging further confirmed that the successful infection of local cells in MDBC by shRNA lentivirus vectors. Meanwhile, the body weights of the rats during the developmental process were recorded, and there was no significant difference between the two groups (S4 Fig). Since the rats were still in the breastfeeding period during the observation, their dietary conditions were not recorded.

Based on the observed morphological changes, downregulated ITGA7 expression levels, and in vivo GFP expression, an effective ITGA7 knockdown animal model was successfully established through local injection of the lentivirus vectors in the suboccipital region.

## Histological changes in MDBC by downregulation of ITGA7 expression during development

Following inhibition of ITGA7 expression in rat MDBC, head and neck samples were collected at P7 and P14 and subjected to sagittal sectioning. Masson staining was utilized to better display the collagen fibers, with muscle fibers staining red and connective tissue fibers staining blue.

At P7, in the shRNA-NC group, the morphology of the MDBC in the dorsal atlanto-occipital interspace exhibited the previously described characteristics [32] (Fig 2A). The RCDmi and SDM were interconnected via MDB, and the RCDmi directly linked to the DAOM, closely fused to the SDM (Fig 2B). The MDB fibers, along with fibrocytes and fibroblasts, demonstrated a well-defined orientation, integrating into the DOAM and SDM at acute angles (Fig 2D, E). Most cells exhibited elongated nuclei aligned with the direction of muscle tension. In contrast, the shRNA-ITGA7 group exhibited a notable increase in the number of fibers between the RCDmi and SDM (Fig 2F). Compared to the shRNA-NC group, the MDB fibers in the shRNA-ITGA7 group appeared disorganized and scattered, with fiber bundles exhibiting varied orientations and loose attachment to the DAOM (Fig 2G). Cells interspersed within these fibers displayed rounder nuclei compared to the shRNA-NC group, and these nuclei were dispersed and lacked directional arrangement (Fig 2I, J). These cells are hypothesized to represent fibroblasts that have yet to fully differentiate into fibrocytes. In comparison to the orderly RCDmi muscle fibers in the shRNA-NC group, the shRNA-ITGA7 group exhibited increased gaps, uneven fiber thickness, partial interruptions of muscle fibers, and elevated levels of interstitium and intramuscular fatty infiltration (Fig 2B, C, G, H). These alterations are reminiscent of the characteristic features of muscle dystrophy.

At P14, in the shRNA-NC group, the MDB fibers originating from the RCDmi formed bundled interconnections with the DAOM and SDM, resulting in tightly fused structures. The number of cells located between these collagen fibers significantly decreased, indicating a progressive maturation of the MDB fibers (Fig 3A, B, D, E). Furthermore,

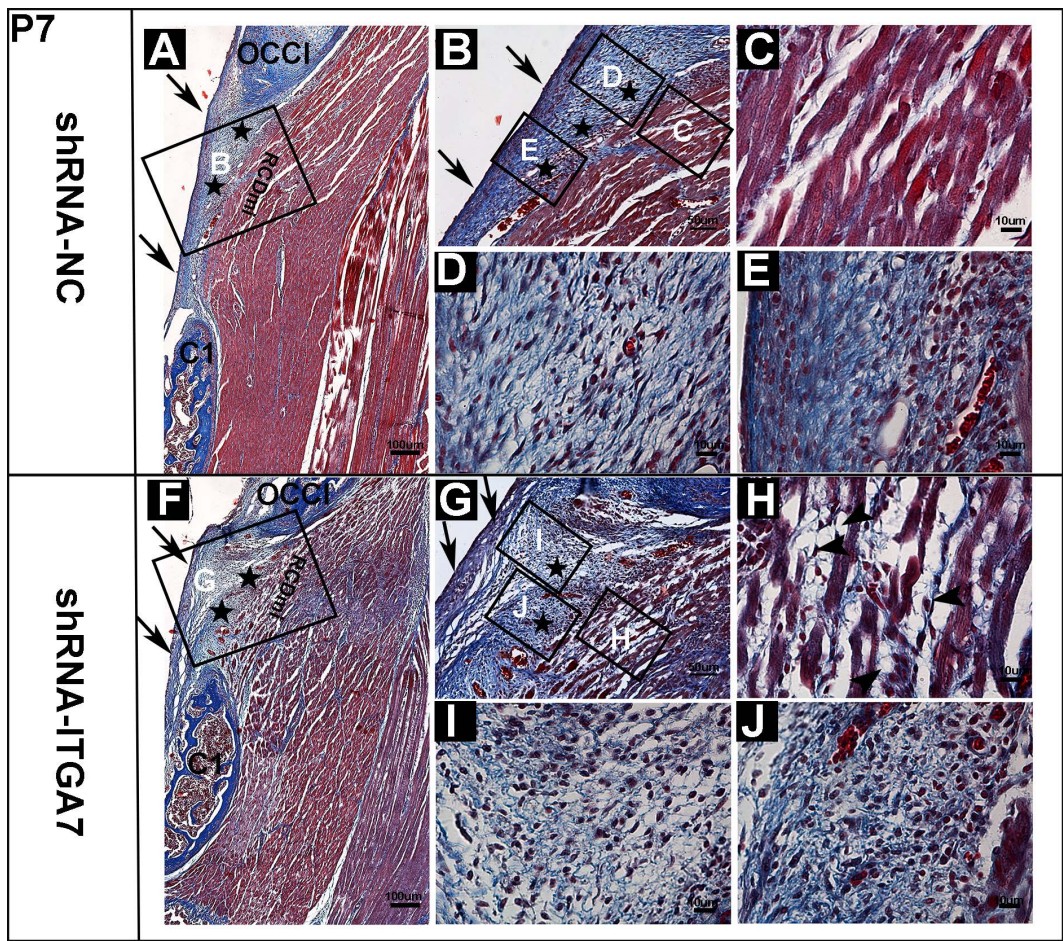

**Fig 2. Masson's trichrome staining of the dorsal atlanto-occipital interspace in rats at P7 after ITGA7 knockdown (n = 4).** Muscle fibers were stained red, and connective tissue fibers blue. (A) A representative image showing Masson's trichrome staining of sagittal section in the shRNA-NC group. (B) Enlarged view of box B in A. (C-E) Enlarged view of boxes in B. (C) The muscle fibers of RCDmi. (D-E) MDB fibers in the shRNA-NC group, MDB fibers, including fibrocytes and fibroblasts, displaying directionality and integration into DOAM and SDM at an acute angle. (F) A representative image depicting Masson's trichrome staining in a sagittal section of the shRNA-ITGA7 group. (G) Enlarged view of box G in F. (H-J) Enlarged view of boxes in G. (H) Increased gaps and adipocytes between RCDmi muscle fibers were observed, black arrowhead indicated adipocytes. (I-J) MDB fibers in the shRNA-ITGA7 group, characterized by a loose, scattered arrangement, with fiber bundles varying in directions and loosely connected to the DOAM. Cells exhibited round nuclei arranged in a dispersed manner, lacking directionality. OCCI: occipital bone. C1: atlas. RCDmi: rectus capitis dorsal minor muscle. ★: MDB fibers; ↑: SDM.

the RCDmi muscle fibers exhibited increased organization and density compared to the shRNA-NC group at P7 (Fig 3A–C). However, in the shRNA-ITGA7 group, MDB fibers displayed an irregular distribution. In the upper region of the atlanto-occipital interspace, a loose connection was observed between the RCDmi and DAOM (Fig 3F). Conversely, in the lower region of the interspace, a significant augmentation of MDB collagen fibers was observed between the RCDmi and SDM. These fibers showed a disordered arrangement, varying from sparse to densely clustered and stacked. MDB fibers still contained a substantial number of fibrocytes and fibroblasts interspersed with visible collagen fibers (Fig 3G, I, J). In addition, a few ovoid fibroblast nuclei lacking directionality persisted between the DAOM and the prominent collagen cluster (Fig 3J). A portion of the RCDmi muscle fibers in the shRNA-ITGA7 group exhibited a morphology like that observed in the shRNA-NC group. However, the lower ventral region still displayed increased gaps and interstitium between muscle fibers (Fig 3H).

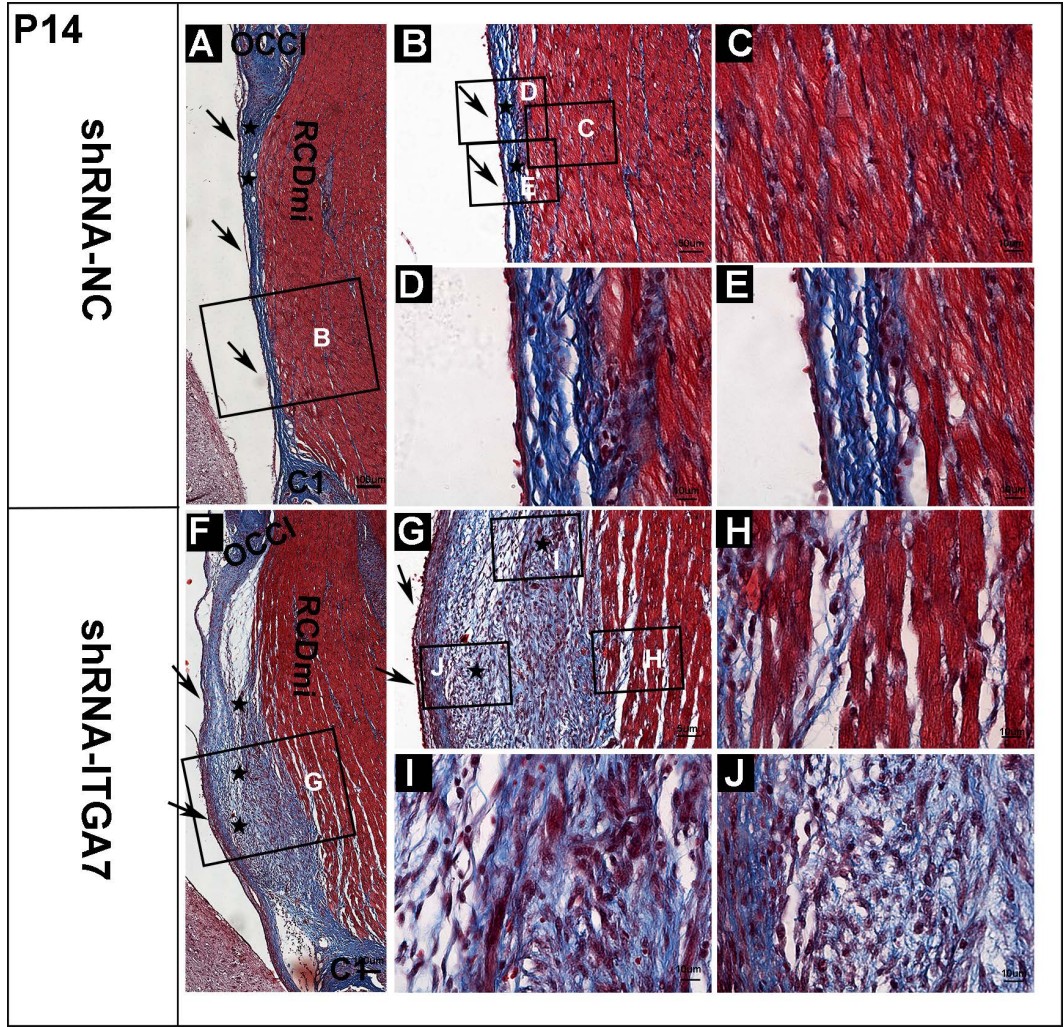

**Fig 3. Masson's trichrome staining of the dorsal atlanto-occipital interspace in rats at P14 after ITGA7 knockdown (n = 4).** (A) A representative image showing Masson's trichrome staining of a sagittal section in the shRNA-NC group. (B) Enlarged view of box B in A. (C-E) Enlarged view of boxes in B. (C) showed the muscle fibers of RCDmi. (D-E) showed the fusion of MDB, DAOM and SDM, characterized by mature cross-linked collagen fibers. (F) A representative image depicting Masson's trichrome staining of a sagittal section in the shRNA-ITGA7 group. (G) Enlarged view of box G in F. (H-J) Enlarged view of boxes in G. (H) exhibited increased gaps between RCDmi muscle fibers. (I) showed immature collagen fibers containing a large number of fibrocytes and fibroblasts. (J) revealed a few ovoid fibroblast nuclei, lacking directionality, situated between the DAOM (left) and the large collagen cluster (right). OCCI: occipital bone. C1: atlas; RCDmi. rectus capitis dorsal minor muscle. ★: MDB fibers. ↑: SDM.

These findings suggest that inhibiting ITGA7 expression impedes development and formation of MDB, manifesting as interrupted the appropriate assembly of MDB fibers, and RCDmi muscle dystrophy.

## Ultra-morphology changes of MDBC of ITGA7 knockdown rats

To further elucidate the morphological changes of MDBC in the ITGA7 knockdown group, the ultra-structures were observed using scanning electron microscopy and transmission electron microscopy.

At P7, under SEM, the MDB fibers in the shRNA-NC group formed bundles of collagen fibers with a relatively uniform orientation that gradually integrated into the dura mater, originating from the ventral side of the RCDmi (Figs 4A, C, D).

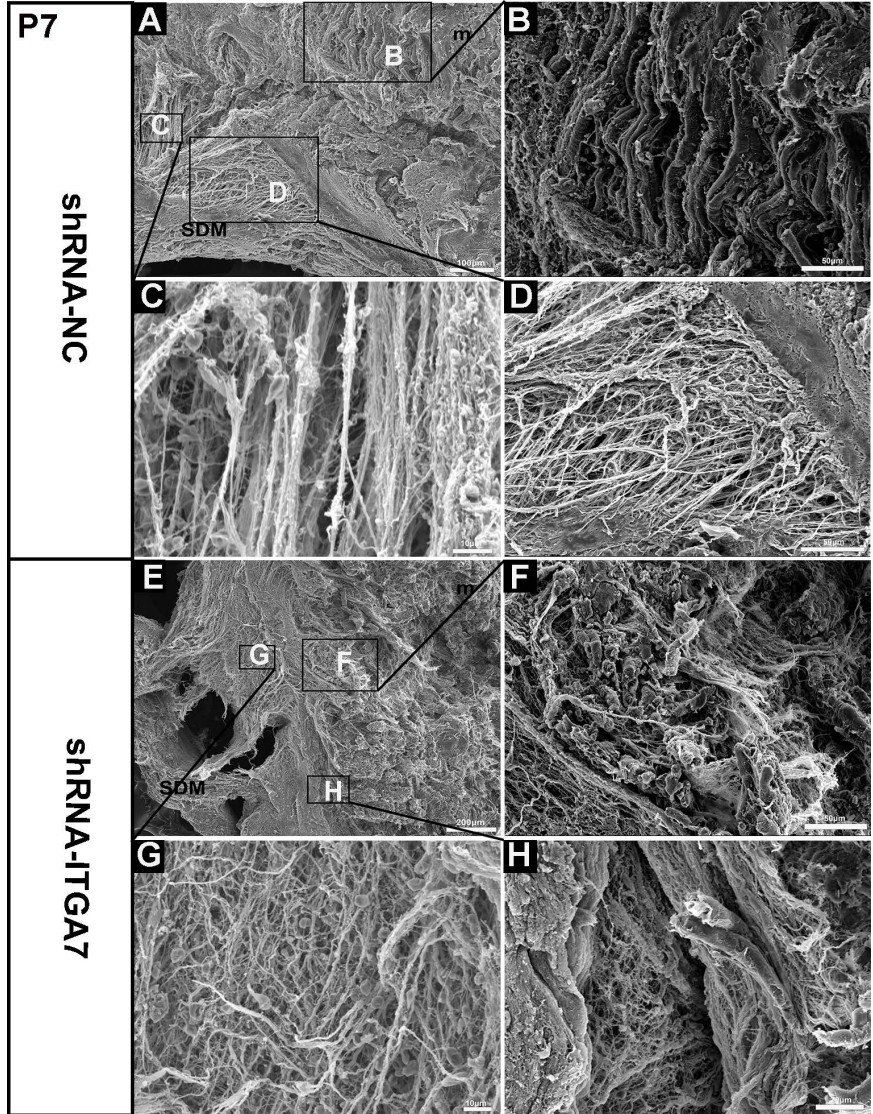

**Fig 4. Scanning electron microscopy of MDBC in the dorsal atlanto-occipital interspace of rats at P7 after ITGA7 knockdown.** (A) A representative image showing a sagittal section in the shRNA-NC group. (B) The muscle fibers of the RCDmi exhibited a regular arrangement. (C) The MDB was composed of collagen fiber bundles arranged in a relatively uniform direction. (D) MDB fibers in the shRNA-NC group oriented from the ventral part of RCDmi and gradually integrated into the SDM. (E) A representative image showing a sagittal section in the shRNA-ITGA7 group. (F) Partial disruption muscle fibers and excessive deposition of interstitial fibers were observed in RCDmi of the shRNA-ITGA7 group. (G) MDB fibers in shRNA-ITGA7 group were composed of fine fibers running in various directions, with fibers bending and overlapping to form a loose network. (H) MDB fibers between RCDmi and SDM in shRNA-ITGA7 group showed a loose network arrangement. ★: MDB fibers. SDM: spinal dura mater. m: rectus capitis dorsal minor muscle.

Concurrently, the muscle fibers in the shRNA-NC group displayed a regular and dense arrangement, indicative of normal muscle development (Fig 4A, B). In contrast, in the shRNA-ITGA7 group, the region typically occupied by dense MDB fibers instead displayed finer fibers oriented in multiple directions and overlapping to form a looser network (Fig 4E, G, H). Additionally, RCDmi muscle fibers displayed partial disruption and an excessive accumulation of interstitial fibers (Fig 4E, F). These findings further confirmed that suppression of ITGA7 expression leads to developmental abnormalities in MDBC.

At P14, under SEM, the MDB fibers in the shRNA-NC group integrated into regular layered DAOM and fused with the SDM in the sagittal section (Fig 5A–C). Compared to P7, the collagen fibers of MDB became denser, thicker, and more regularly arranged (Fig 5C). Conversely, in the shRNA-ITGA7 group, although collagen fibers showed gradual maturation relative to P7, their arrangement remained disordered, with fibers of varying thicknesses clustered in different directions (Fig 5E, F). Under TEM, MDB fibers in the shRNA-NC group formed bundles with relatively consistent orientations. A few fibrous cells were interspersed among the MDB fibers (Fig 5H, I). Conversely, in the shRNA-ITGA7 group, collagen fibers were irregularly arranged and crisscrossed, varying in size and forming relatively small bundles. Numerous fibroblasts,

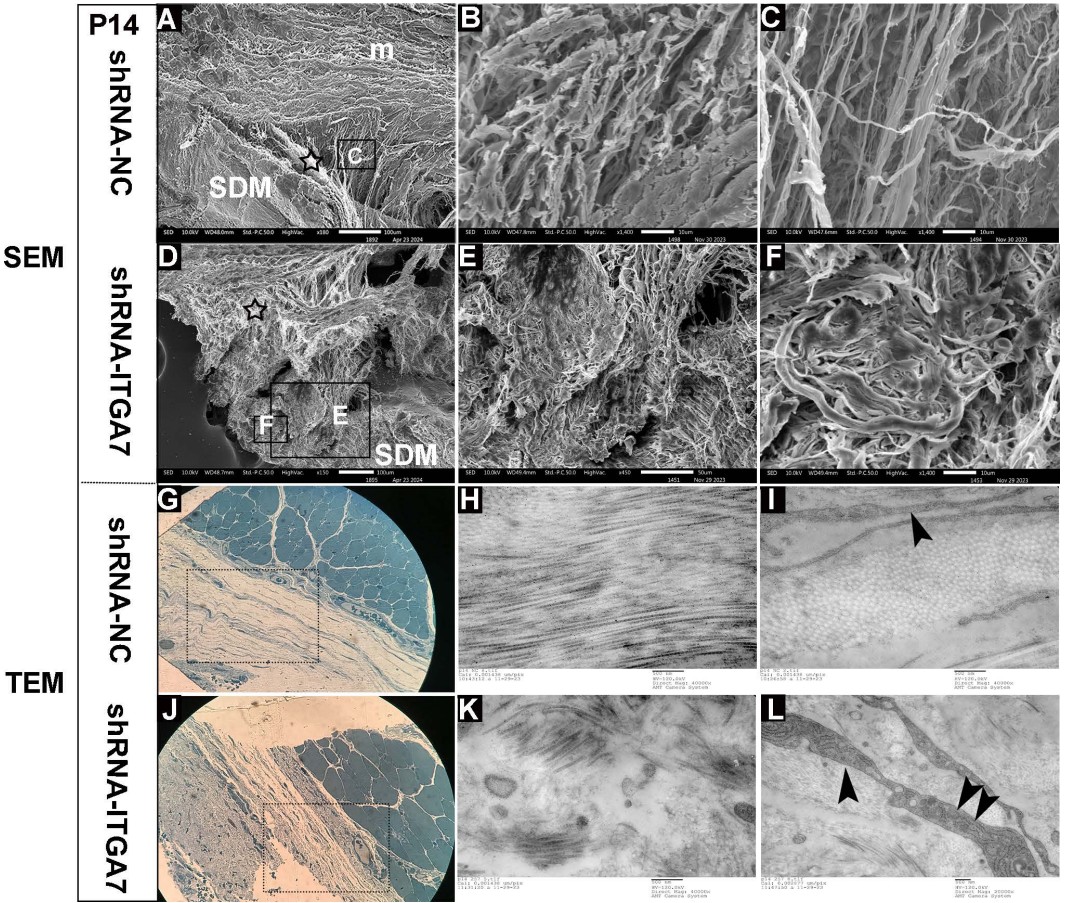

**Fig 5. Scanning and transmission electric microscopy of the atlanto-occipital interspace in rats at P14.** (A-F) were photographed using a scanning electron microscope, (G-L) were captured using a transmission electric microscope. (A) A representative image showing a sagittal section in the shRNA-NC group. (B) Enlarged view of MDB fibers in the shRNA-NC group. (C) MDB fibers in the shRNA-NC group were formed by regular arranged mature collagen fibers. (D) A representative image showing a sagittal section in the shRNA-ITGA7 group. (E) Enlarged view of MDB fibers in the shRNA-ITGA7 group. (F) MDB fibers in the shRNA-ITGA7 group showed a disordered arrangement, with fibers of varying thicknesses clustered and in multiple directions. (G) A representative image of Toluidine blue staining semi-thin section in the shRNA-NC group. (H) MDB fibers in the shRNA-NC group exhibited bundles of collagen fibers aligned in relatively consistent directions. (I) MDB cross-sections consisted of bundles of collagen fibers with relatively uniform thickness in the shRNA-NC group, black arrowhead showed a fibrocyte in MDB. (J) A representative image of Toluidine blue staining semi-thin section in the shRNA-ITGA7 group. (K) Collagen fibers in the shRNA-ITGA7 group appeared irregularly arranged and crisscrossed, with varying sizes and smaller bundles. (L) Numerous fibroblasts, rich in rough endoplasmic reticulum, were observed among the disordered MDB fibers in shRNA-ITGA7 group, black arrowheads showed the rough endoplasmic reticulum. SDM: spinal dura mater. ★: MDB fibers. m: rectus capitis dorsal minor muscle.

characterize by abundant rough endoplasmic reticulum, were observed interspersed between the fibers (Fig 5K, L), suggesting that the collagen fibers have not yet fully matured.

Taken together, these findings demonstrate that inhibiting ITGA7 expression results in developmental abnormalities in both MDB fibers and RCDmi within the MDBC. Abnormalities in Collagen fiber arrangement and thickness were more distinctly observed under SEM. TEM revealed a disordered arrangement of collagen fibers and an abundance of fibroblasts, indicative of immature collagen fibers differentiation.

### Inhibition of ITGA7 changed collagen volume fraction and fiber properties of MDB

Masson staining revealed an increased abundance of collagen fibers in the MDBC region of the shRNA-ITGA7 group compared to the shRNA-NC group. Semi-quantitative analysis was applied to assess the collagen volume fraction (CVF) of the MDBC at P7 and P14, respectively. The CVF in the shRNA-ITGA7 group was significantly higher than in the shRNA-NC group at both P7 and P14 (* $p < 0.05$) (S5A–C Fig, S3 Table). These findings suggest that downregulation of ITGA7 expression is associated with an increase in the amount of collagen fibers within the dorsal atlanto-occipital interspace.

To characterize the collagen properties of these abundant fibers in the atlanto-occipital interspace in the shRNA-ITGA7 group, Picrosirius red staining was utilized. Under a polarizing microscope, the fibers of the MDB, DAOM and SDM in the shRNA-NC group at P7 presented a birefringence of yellow to red color, suggesting a predominance of collagen type I. As development progressed, these structures exhibited increasingly intense red refraction, which was more pronounced by P14 (S6A, A', C, C' Fig). Although Masson staining indicated a greater density of fibers in the shRNA-ITGA7 group, Picrosirius red staining demonstrated that these fibers exhibited weak yellow birefringence and a more pronounced presence of greenish collagen type III at both P7 and P14 (S6B, B', D, D' Fig). At both P7 and P14, the proportion of orange-red fibers in Sirius red staining was significantly higher in the shRNA-ITGA7 group than in the shRNA-NC group (* $p < 0.05$) (S6E Fig, S3 Table). The abundant fibers either lacked birefrigence or displayed weak yellow birefrigence, suggesting they were composed primarily of immature collagen fibers. These results further substantiate that the maturation of MDB fibers is impeded by the suppression of ITGA7 expression.

### Inhibition of ITGA7 induced muscle atrophy and a decline in muscle force in RCDmi

To further verify the impact of ITGA7 knockdown on RCDmi muscle force, cross-sections of RCDmi were performed and stained with laminin to show the outlines of muscle fibers. Subsequently, the cross-sectional area of muscle fibers was calculated and analyzed in each group. At P7, the RCDmi. muscle atrophy was observed in shRNA-ITGA7 group, as evidenced by decreased myofiber cross-sectional area (Fig 6A–C). However, no significant change in muscle atrophy was observed in shRNA-ITGA7 group at P14 (Fig 6A–C).

The maximum breaking force of muscle serves as an indicator of muscle force. The uniaxial tensile experiments were conducted to evaluate the muscle force of RCDmi in each group (S2 Fig). At P7, the maximum breaking force of RCDmi of the shRNA-ITGA7 group was significantly lower than that of the shRNA-NC group. At P14, there was no significant difference between the two groups (Fig 6D, S3 Table).

Therefore, it can be concluded that the knockdown of ITGA7 resulted in a decrease in muscle force within the RCDmi.

### Inhibition of ITGA7 altered ECM related genes expression in RCDmi

To further investigate the mechanisms of the increase in fiber content and the occurrence of muscle dystrophy in the shRNA-ITGA7 group at P7, the mRNA expression levels of ECM deposition and ligament formation related genes were detected. Moreover, a fibro-adipogenic progenitors(FAPs) marker gene Pdgfra was also evaluated in the two groups. The expression levels of Col1a2, Lama2 were decreased in the shRNA-ITGA7 group than in the shRNA-NC group.

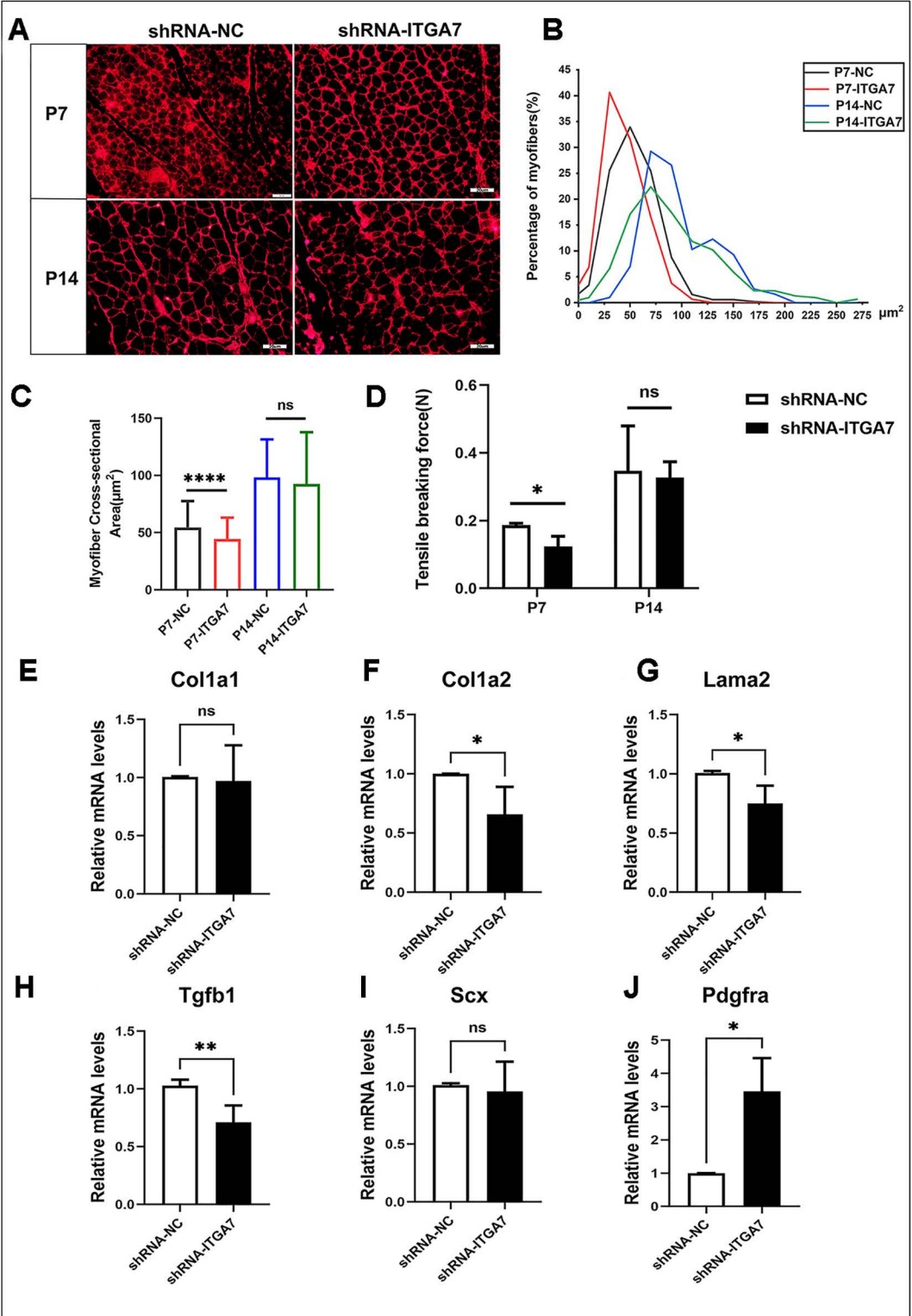

**Fig 6. Inhibiting ITGA7 reduced muscle force and altered ECM related genes expression.** (A) Expression of Laminin α2 in the RCDmi in rats. (B) The proportion of the cross-sectional area of the RCDmi in shRNA-NC and shRNA-ITGA7 at P7 and P14. (C) The average cross-sectional area of the RCDmi in shRNA-NC and shRNA-ITGA7 at P7 and P14. (D) The maximum breaking force of the RCDmi in shRNA-NC and shRNA-ITGA7 groups at P7 and P14. (E) The mRNA expression levels of Col1a1 in the MDBC of rats in the atlanto-occipital interspace. (F) The mRNA expression levels of Col1a2

in the MDBC of rats in the atlanto-occipital interspace. (G) The mRNA expression levels of Lama2 in the MDBC of rats in the atlanto-occipital interspace. (H) The mRNA expression levels of Tgfb1 in the MDBC of rats in the atlanto-occipital interspace. (I) The mRNA expression levels of Scx in the MDBC of rats in the atlanto-occipital interspace. (J) The mRNA expression levels of Pdgfra in the MDBC of rats in the atlanto-occipital interspace. (n = 4, * $p < 0.05$, ** $p < 0.01$).

However, the expression levels of Col1a1 showed no difference between the two groups (Fig 6E–G, S3 Table). The expression levels of Tgfb-1 in the shRNA-ITGA7 group were decreased than those in the shRNA-NC group (Fig 6H, S3 Table). We speculate that it might be due to the decreased expression of ITGA7, thus the reduction of mechanical forces from muscle, which leads to the insufficient activation of Tgfb-1, and further affects the formation of MDB fibers. As the key transcription factor of tendon and ligament formation, the expression levels of scleraxis (Scx) showed no significantly difference between two group (Fig 6I, S3 Table). Meanwhile, there was a increase in the expression of Pdgfra in shRNA-ITGA7 group (Fig 6J, S3 Table), which indicated the activation of FAPs, this could explain the abnormally accumulation of interstitial fibers and intramuscular fat infiltration in histology.

## Discussion

Previous studies have established that head movements facilitate CSF circulation by transferring tension from the suboccipital muscles to the SDM via the MDB [11,28,44]. This study investigated the role of ITGA7 in MDB development by establishing a model of local ITGA7 knockdown in the atlanto-occipital interspace of rats. Inhibtion ITGA7 expression impedes development and maturation of MDB, manifesting as disrupted the MDB fibers appropriate assembly, RCDmi muscle dystrophy, and reduced muscle force. A key finding of our study is the importance of ITGA7 as a direct molecular link between suboccipital muscles and MDB. This indicate that MDB fiber formation and maturation are dependent on the force generated by suboccipital muscles.

ITGA7 is an extracellular matrix-binding protein. Recent studies on ITGA7 includes its effect on muscle stem cells polarity [45], adult-onset cardiomyopathy [46], and tumor progression, prognosis [47]. While its roles in skeletal muscle development and force transmission have been extensively studied previously [33,39], this study focused on the local effects of ITGA7 modulation. Local injection was used to regulate ITGA7 expression, minimizing systemic impact. Our data demonstrated that significant morphological abnormalities in MDBC after interfering with ITGA7 expression. Increased gaps and expanded muscular interstitium were noted in the muscle fibers of RCDmi, accompanied by adipocyte infiltration at P7. Consistent with previously study, ITGA7 serves as a 'bridge' connecting muscle fibers to the ECM, and its deficiency leads to progressive muscular dystrophy [40]. When skeletal muscle homeostasis is disrupted, fibro-adipogenic progenitors may progress towards fibrosis or adipogenesis [48]. The observed muscle fibrosis and fat infiltration in RCDmi were likely due to activated fibro-adipogenic progenitors, supported by the increased expression of Pdgfra in shRNA-ITGA7 group. Additionally, the observed improvement in muscle dystrophy at P14 may be attributed to the limited efficiency of *in vivo* lentivirus infection, resulting in a reduced proportion of infected cells during development, and the regenerative capacity of muscle cells. Meanwhile, in the region where MDB should formed, redundant, heterogeneously oriented fibers were observed in the dorsal atlanto-occipital interspace. Cells in this region displayed rounder nuclei, and lacked directional arrangement. ITGA7 deficiency impaired muscle fiber attachment to the ECM and altered force transmission between myocytes and the ECM [49]. Cells can sense mechanical stimuli and convert these into biochemical signals, influencing cell and tissue morphology, composition and physiological functions [50]. Reducing ITGA7 expression in RCDmi likely diminishes force transmission between RCDmi and MDB forming cells in the dorsal atlanto-occipital interspace. The observed decrease in muscle force resulted from both weakened direct RCDmi-ECM connections and muscle atrophy. The increased number of fibers observed may represent a compensatory mechanism. At P14, the MDB fibers matured in the shRNA-NC group, whereas a developmental delay was observed in the shRNA-ITGA7 group. TEM revealed irregular arranged MDB fibers and numerous fibroblasts between them, suggesting immaturity. Picrosirius red

staining confirmed this developmental delay, showing increased immature collagen type I and greenish collagen type III fibers. These findings suggest that ITGA7 inhibition results in improper MDB fiber formation and developmental delay.

Taken together, these results demonstrate that inhibiting ITGA7 expression disrupts mechanical force transmission form RCDmi to MDB-forming cells leading to MDBC developmental abnormalities. This aligns with existing knowledge regarding the importance of mechanical forces for myotendinous junction maturation and tendon cell differentiation [51,52]. Under mechanical forces, fibroblasts secrete collagen to help tendons resist the load generated by muscle contraction. During tendon differentiation, muscle-exerted force on the ECM is crucial for activating transforming growth factor beta (Tgf-β) at the muscle-tendon interface, promoting Tgf-β binding and downstream activation of SCX, which induces collagen synthesis and deposition [52,53]. ITGA7 inhibition likely impairs this pathway, resulting in decreased Tgf-β expression. The force generated by the developing RCDmi muscle is therefore essential for MDB fiber differentiation and maturation, confirming that MDB development depends on the force exerted by the suboccipital muscles. This study suggests that suboccipital muscle force transmitted to the SDM via the MDB may act as a potential dynamic driver of CSF circulation.

Abnormal MDBC development due to ITGA7 deletion may affect CSF circulation. Numerous studies suggest that the MDB may couple suboccipital muscle movements to cerebrospinal fluid (CSF) dynamics in humans [11,44] and animal models (dogs [54], alligators [24,55,56], rats [57]). For example, head movements (flexion-extension, rotation) and suboccipital muscle stimulation via MDB integrity increase CSF pressure, while MDB disruption abolishes this effect. Rat models further show that suboccipital muscle hypertrophy/atrophy correlates with altered CSF secretion and reabsorption rates, linking MDB-mediated mechanical forces to CSF circulation regulation [8]. Bleomycin-induced MDB and fibrous tissue hyperplasia in the atlanto-occipital interspace can decrease dura mater compliance, negatively impacting CSF circulation [58]. The occipito-atlantal cistern (OAC), a subarachnoid cistern extending from the foramen magnum to the upper edge of the axis, is a key region for MDB action [59]. In the shRNA-ITGA7 group, excessive MDB fibers accumulation may ventrally compress the SDM, potentially reducing OAC volume. It is plausible that this condition could have effects analogous to cerebellar tonsillar herniation (a disorder associated with CSF circulation disruption), though such a link remains speculative and requires direct investigation. Moreover, pathological MDBC alterations have also been linked to chronic headaches. Hack [60], et al. found that headache symptoms significantly relieved after surgical transection of the connection between the RCPmi and the dura mater in chronic headache patients. Yuan [27], et al. demonstrated that the long-axis cross-sectional area of the RCDmi was significantly larger in chronic headache patients compared with healthy volunteers. Uthaikhup S [61], et al. found muscle atrophy with increased fatty infiltration in the rectus capitis posterior major and minor in older women with cervicogenic headache. These findings suggest a potential role of the MDBC in headache. During atlanto-occipital surgeries, avoiding excessive MDB damage and reinforcing MDB-dura connections via suture techniques are critical. This may help preserve mechanical buffering, support CSF circulation, and reduce potential risks of headache and CSF leakage related to CSF dynamic disorders [62], though clinical confirmation is needed. Inhibiting ITGA7 in the MDBC results in abnormal development. Future research should focus on investigating the clinical implications of such abnormal MDBC development, especially its potential association with headaches.

While this study provides important insights into MDB development, several limitations should be considered. First, the translational relevance of rodent findings to human physiology requires further validation, given anatomical and biomechanical differences in craniocervical junction structures across species. Second, our focus on ITGA7 does not preclude potential compensatory or synergistic effects from other ECM components (e.g., Col1a1, Lama1, Vimentin). The complex interactions between multiple ECM proteins in MDB formation warrant systematic investigation in future studies. Third, the use of mixed-sex animals in our experiments prevents us from assessing potential sex-based differences in the role of ITGA7 in MDB development. Therefore, in future studies, multi-omics studies will be crucial for identifying the regulation genes and pathways, and ultimately providing a more complete understanding of the molecular mechanisms underlying MDB development.

## Conclusion

This study successfully established a rat model of locally ITGA7 inhibition in the suboccipital region. Reduced ITGA7 expression in RCDmi led to MDBC developmental abnormalities, highlighting the role of ITGA7 as a key regulator in MDBC development. This study provides compelling evidence that MDB differentiation and maturation depend on the mechanical force generated by the suboccipital muscles.

## Supporting information

**S1 Fig. MDBC sampling process: E16 and P7 stage sample collection.** (A) Gross view of the E16 embryo tissue specimen, with the black box marking the area for locating key anatomical structures. (B) Magnified view of the boxed area in Figure A, for assisting in the accurate identification of MDBC – related structures at the embryonic stage. The dotted circles represent C1, C2, and OCCI, respectively. (C) The area from the inferior margin of the occipital bone to the atlas is the MDBC sampling site at the E16 stage. (D) Neck of a P7 neonatal rat before surgery. (E) Surgical incision made on the neck. (F) Superficial muscles dissected to expose the rectus capitis dorsalis major muscle, allowing clear visualization of the occipital bone and atlas. (G) The tissue in the atlanto – occipital space is dissected from the cephalic to the caudal direction. (H) Demonstration of placing the tissue back to its original position. OCCI: occipital bone. C1: atlas. C2:axis.
(TIF)

**S2 Fig. Pictures of the biomechanical tests on the RCDmi.** (A) Biomechanical Testing Device with 20 N capacity load cell. (B) Fixtures in Biomechanical Testing Device. (C) Exposed the RCDma and RCDmi. (D) Apply bandages to the cranial and caudal sides of the RCDmi and then place it in the fixture to perform uniaxial stretching.
(TIF)

**S3 Fig. The expression of Itga7 in the MDBC of Rats at different developmental stages.** Itga7 expression in rats at four critical stages: embryonic day 16 (E16), postnatal day 0 (P0), postnatal day 7 (P7), and postnatal day 14 (P14) were evaluated by qRT-PCR. The expression of Itga7 increased progressively during MDBC development (*$p < 0.05$, **$p < 0.01$).
(TIF)

**S4 Fig. The weight of rats after localized ITGA7 knockdown.** The body weights of the rats during the developmental process showed no significant difference between the two groups ($p > 0.05$).
(TIF)

**S5 Fig. Collagen volume fraction in the MDBC of the dorasl atlanto-occipital interspace after localized ITGA7 knockdown.** (A) Masson's trichrome staining of the dorsal atlanto-occipital interspace in rats at P7 and P14 after localized ITGA7 knockdown. (B) The fibers of the dorsal atlanto-occipital interspace are circled in yellow by Image J software. The area of fibers are significantly increased in the shRNA-ITGA7 group. (C) Comparison of the area of the fibers between the shRNA-NC group and the shRNA-ITGA7 group. (*$p < .05$; ns: no statistical significance).
(TIF)

**S6 Fig. Sirius Red staining of the dorsal atlanto-occipital interspace in rats at P7 and P14 after localized ITGA7 knockdown.** (A-D) were captured using a normal light microscope. (A'-D') represent the same section observed under a polarized light microscope. (A and A', B and B') corresponded to the shRNA-NC and shRNA-ITGA7 group at P7, respectively. (C and C', D and D') corresponded to the shRNA-NC and shRNA-ITGA7 group at P14, respectively. (A') MDB fibers in the shRNA-NC group presented birefringence ranging form yellow to red. (C') The red birefringence of MDB fibers was intensified at P14. (B' and D') MDB fibers in the shRNA-ITGA7 group displayed a weak yellow birefringence, with an increase greenish collagen type III. (E) Proportion of orange – red fibers from Sirius Red staining between the shRNA

- NC group and the shRNA - ITGA7 group. (* p < 0.05) All image acquisition parameters, including exposure time (150 ms) and light sensitivity (ISO 200) were consistent. RCDmi: rectus capitis dorsal minor muscle. ★: MDB fiber. ↑: SDM.
(TIF)

**S1 Table. The target sequences for stable knockdown of ITGA7.**
(DOCX)

**S2 Table. Primer sequences used.**
(DOCX)

**S3 Table  Summary table of main quantitative results.**
(DOCX)

## Author contributions

**Conceptualization:** Yun-Feng Liu, Wei Ma.

**Data curation:** Lu Zhang, Yun-Feng Liu, Yan-Yan Chi.

**Formal analysis:** Lu Zhang, Yun-Feng Liu, Mi Luo, Xue Song, Xin-Yuan Zhang, Xiao-Ying Yuan, Ruo-Tong Zhang.

**Funding acquisition:** Xiao-Ying Yuan, Ruo-Tong Zhang, Hong-Jin Sui.

**Investigation:** Lu Zhang, Yun-Feng Liu, Mi Luo, Xue Song, Xin-Yuan Zhang, Xiao-Ying Yuan.

**Methodology:** Lu Zhang, Yun-Feng Liu, Mi Luo, Xue Song, Jing-Xian Sun, Jian-Fei Zhang, Wei Ma.

**Software:** Lu Zhang, Yun-Feng Liu, Mi Luo, Xue Song.

**Supervision:** Xue Song, Jing-Xian Sun, Jian-Fei Zhang, Sheng-Bo Yu, Wei Ma, Hong-Jin Sui.

**Validation:** Lu Zhang, Yun-Feng Liu.

**Writing – original draft:** Lu Zhang, Yun-Feng Liu.

**Writing – review & editing:** Chan Li, Campbell Gilmore, Wei Ma, Hong-Jin Sui.

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
