## [Decision Letter · Decision Letter 0]

22 Apr 2025

Dear Dr. Ma,

Thank you for submitting your manuscript to PLOS ONE. After careful consideration, we feel that it has merit but does not fully meet PLOS ONE’s publication criteria as it currently stands. Therefore, we invite you to submit a revised version of the manuscript that addresses the points raised during the review process.

Authors are requested to submit a revised manuscript addressing all reviewers' comments reported below and in the attached file. Macroscopic evidence of MDB dissection and evidence of compliance with the Arrive guidelines should be provided. The fact that a transcriptome paragraph is reported in the M&M section but the transcriptome data are not reported in the Results section is not acceptable. The sentence reported in the Results section "Differentially expressed genes (DEGs) associated with MDBC development were identified (unpublished data, not detailed in this study)" means that the authors have performed transcriptome analyses, they partially use the transcriptome results to corroborate the reported data, but they did not want to "officially" report them in the Results section. This seems to me to be ethically unacceptable.

We look forward to receiving your revised manuscript.

Kind regards,

Aldo Corriero, Ph.D.

Academic Editor

PLOS ONE

Additional Editor Comments (if provided):

Reviewers' comments:

Reviewer's Responses to Questions

**Comments to the Author**

1. Is the manuscript technically sound, and do the data support the conclusions?

Reviewer #1: Yes

Reviewer #2: Yes

Reviewer #3: Partly

2. Has the statistical analysis been performed appropriately and rigorously?

Reviewer #1: No

Reviewer #2: Yes

Reviewer #3: Yes

3. Have the authors made all data underlying the findings in their manuscript fully available?

Reviewer #1: Yes

Reviewer #2: Yes

Reviewer #3: Yes

4. Is the manuscript presented in an intelligible fashion and written in standard English?

Reviewer #1: No

Reviewer #2: Yes

Reviewer #3: Yes

Reviewer #1: The study presents novel findings and a robust experimental approach, contributing to our understanding of MDB development. However, before acceptance, the following key revisions are necessary:

Introduction: Reorganize paragraphs to enhance the logical flow of ideas. Eliminate unnecessary repetitions, e.g.: - Cerebrospinal fluid (CSF) circulation… that maintains CSF homeostasis.

-The MDB consists of fibrous connective tissues that runs between the suboccipital muscles and the cervical spinal dura mater (SDM)...

-The human MDB is a tendinous-like structure primarily composed of parallel arranged collagen type I fibers...

-The MDB, like tendons, is primarily composed of strong collagen fibril arrays ..., the example about CSF circulation and MDB composition.

-Comparative anatomical studies have demonstrated that the MDB is a highly conserved structure with significant physiological functions in mammals, reptiles and birds...

-It is speculated that mechanical stress from suboccipital muscles may contribute to MDB maturation...

-Ensure a clear progression from the general description of the MDB to specific details.

MM: Include a power analysis to justify the sample sizes used (n=3 or n=4). Add an extra untreated control group to rule out non-specific effects from the injection procedure. Perform quantitative analyses of collagen fiber density or optical density measurements from Picrosirius Red and Masson staining images.

R: Conduct a functional enrichment analysis using Gene Ontology (GO) or KEGG pathway to identify other relevant genes. This will strengthen the conclusions about molecular players in MDB development.

Discussion: Discuss the relevance of these additional genesAddress potential limitations, such as extrapolating findings from rats to humans and possible interactions with other extracellular matrix proteins that might influence MDB development.

Provide a more detailed discussion on the potential clinical implications of the findings, including the role of the MDB in CSF circulation and its relationship to cervicogenic headaches.

Reviewer #2: Initially, I received the first version of the manuscript where the authors claimed compliance with the ARRIVE guidelines. However, it was evident that this was not the case. After requesting clarification from the editor, a revised version was provided. Despite improvements, I still have doubts about the complete adherence to the ARRIVE guidelines. While the research conducted is both interesting and credible offering significant insights into the development of the myodural bridge (MDB), I suggest that the authors submit another revision. This revision should thoroughly address all items of the ARRIVE guidelines explicitly indicating where each is resolved within the manuscript. For example, it is not sufficiently clear how the sample size for each group was calculated. Additionally, as a minor suggestion, the sentence "Future research should investigate the clinical implications of abnormal MDBC development, particularly in relation to headaches." should be removed from the conclusions section. The authors might consider relocating it to the Discussion section, as it is not directly related to the findings of the current study.

Reviewer #3: General.- The manuscript entitled “Force-dependent development of the myodural bridge in rats:the impact of Integrin α�” by Lu Zhang and co-workers is an interesting contribution to the knowledge of a relatively unknown structure such as the myodural bridges (MDB).

MDB are specialized fibrous structures that connect the suboccipital musculature and ligaments (f.e. septum nuchae) and the spinal dura mater (SDM) at different levels, especially the atlanto-occipital and atlanto-axial spaces. Together these structures forms the myodural bridge complex (MDBC). This complex is regarded as an evolutionarily conserved anatomical structure present in different vertebrate species including terrestrial and marine mammals, birds and reptilians, and also humans. Structurally, MDBC consists of connective tissue containing vessels and scarce nerves; recently sensory nerve formations (presumably related to proprioception) have been described in human MDBC. Functionally, is theorized that MDBC stabilize the dura mater during the extension of the head and neck, thus serving as dural tension monitor, preventing compression of the dura mater during motion of the spinal column and infolding of the dura mater and disruption of the flow of cerebrospinal fluid.

This research, carried out in rats, explores the development of MDBs and their capacity as a transmitter of forces. The manuscript is well written, the objectives are clear, and the techniques are appropriate. The authors identify Integrin α7 (ITGA7) as a key molecular mediator in the development of MDBs, which plays a critical role in the transmission of muscle force. The different experimental models show that the formation of MDBs depends on the suboccipital muscles, suggesting that mechanical forces from suboccipital musculature is fundamental to MDB differentiation and maturation.

Despite the interest of the study, in the opinion of this reviewer some corrections and revisions should be made before it can be published, which are included in the following lines.

Abstract.- It should be clear that there is no one MDB but that almost always there are several, which depends on the species; In addition, fibrous connections are not only established between the duramater and the sub-occipital muscles, but also with the suboccipital ligaments. This is valid for the entire manuscript.

The expression "to cerebrospinal fluid dynamics" should also be modified as it can be confusing. MDBS does not participate in the dynamics of cerebrospinal fluid dynamics but releases the upper cervical leptomeningeal space through which the cerebrospinal fluid circulates.

Finally, the last sentence of the Abstract is not based on the results. It should be eliminated "... and its contribution to cerebrospinal fluid circulation dynamics" because this aspect has not been addressed in the present research.

Introduction.- The information it contains is correct and well structured. The objectives of the work are well specified. Nevertheless, it should not focus on the circulation of the cerebrospinal fluid but on the MDB and/or MDBC, clarifying that these are two anatomically distinct entities: the first are bands of fibrous tissue, the second includes muscles and ligaments.

Materials and methods.- The authors specify Ethical Statements, animal models and study techniques. The techniques are described in sufficient detail to allow replication of the experiments.

Results.- I consider that before molecular biology studies the authors should show two aspects: the dissection and macroscopic appearance of the regions studied and their histology. In fetuses and embryos that are also small, in which the size of the MDBs is extremely small, it is important that the experiments have been carried out on the appropriate tissues and not on other nearby ones. Once it has been demonstrated that it is really about the MDBs, the rest of the studies carried out seem to be adequate, and the results obtained credible and evident.

On the other hand, it is worth highlighting the high quality of the structure, ultrastructure and immunohistochemical images that illustrate the results.

Discussion.- It's correct

Conclusion.- There is correct and unlike the conclusions of the abstract it does not refer to the circulation of the cerebrospinal fluid.

**Do you want your identity to be public for this peer review?** For information about this choice, including consent withdrawal, please see our Privacy Policy

Reviewer #1: No

Reviewer #2: No

Reviewer #3: No

---

## [Author Response · Author response to Decision Letter 1]

16 Jun 2025

Dear Editors and Reviewers,

Thank you for your meticulous review and valuable suggestions on our manuscript! We have carefully considered your comments, and the responses to the revision suggestions are as follows:

1.The macroscopic evidence of MDB dissection is provided in the Supplementary Materials. The evidence of compliance with the ARRIVE guidelines is reflected in the revised manuscript.

2.Regarding the issue of transcriptome data: First and foremost, we would like to express our sincere gratitude to the Editors and Reviewers for the precious advice on our manuscript! We sincerely apologize for any inadequacies in our initial handling of the transcriptome data analysis. Our team has thoroughly discussed the comment of “adding more content related to transcriptome data”. After careful consideration in combination with the subsequent research plan, we have decided, following repeated and prudent deliberation, to remove this part of the content from this article. We sincerely hope for your understanding and support. The explanations are as follows:

(1) The transcriptome data are not the core content of this study, and their removal does not affect the integrity of this article:

The core objective of this study is to explore the role of the target gene ITGA7 in the development process of MDBC, and to elucidate the force - dependent developmental mechanism of the myodural bridge (MDB). In the preliminary exploration stage, through literature research, it was found that the Integrins, particularly ITGA7, are critical for skeletal muscle attachment to connective tissues, and their function is involved in the transmission of lateral and longitudinal forces in skeletal muscle. Moreover, the development of MDB itself is inseparable from force transmission. Given the force - related characteristics of ITGA7 and the dependence of MDB development on force transmission, we initially speculated that ITGA7 might be involved in MDB development. Subsequently, we carried out the mRNA expressions of itga7 at the key developmental stages of MDBC in rats. The results showed that ITGA7 exhibited significant differential expression at different development stages. Even after excluding the transcriptome-related content, the rationale for this study on ITGA7 can still be elucidated. The above statement can fully explain the starting point of this research without relying on the detailed data from transcriptome sequencing in mice.

(2) Research on transcriptomics is ongoing, and detailed results will be published soon:

Currently, our research team is performing in-depth analyses and mining of transcriptome data related to MDB development, alongside experimental validation. Subsequently, we will also upload the complete data to the GEO database. Therefore, elaborating on this in the current article would lead to redundant content and deviate from the current research focus.

(3) Specific adjustment plan for removing the transcriptome content:

To ensure the logical coherence of the article, we will make the following modifications:

Introduction section: Delete the original paragraph related to the transcriptome.

Materials and Methods section: Delete the method for transcriptome sequencing.

Results section: The subtitle “Transcriptome sequencing reveals the potential value of ITGA7 in the developmental regulation of MDBC” is deleted.

Discussion section: Add explanations regarding the limitations of the transcriptomic data, for example:

“A more comprehensive and in - depth exploration of the transcriptomic data related to MDB development is still lacking. Therefore, in future studies, in - depth mining of such data will be crucial for identifying the regulation genes and signaling pathways, and ultimately providing a more complete understanding of the molecular mechanisms underlying MDB development.”

Thank you again for your professional advice! If you think that some transcriptome information still needs to be retained, we can provide the relevant content in the Supplementary Materials for readers' reference.

Thank you again for your professional comments! If there are any other suggestions, we will fully cooperate with the revisions.

Best regards,

Wei Ma

Hong-jin Sui

Reviewer #1:

Introduction

1.The content regarding CSF circulation has been removed, and repetitive descriptions of MDB composition have been revised.

We have carefully reviewed the text and reorganized the paragraphs to ensure a more coherent progression of ideas. Specifically, we have eliminated the redundant descriptions of cerebrospinal fluid (CSF) circulation, removing overlapping statements about its role in maintaining CSF homeostasis. Similarly, we have streamlined the descriptions of the MDB's composition. Instead of separately stating that “the human MDB is a tendinous-like structure with parallel arranged collagen type I fibers” and later repeating that “the MDB, like tendons, is primarily composed of strong collagen fibril arrays”, we have combined these points into a single, concise description. This approach not only enhances readability but also makes the text more efficient in conveying key information about the MDB. We believe these changes have significantly improved the quality of the Introduction, and we appreciate your guidance in helping us refine our manuscript.

2.Revised Logical Flow: From General MDB Characterization to Specific Structural Details

We reorganized the content in a hierarchical manner. First, we began with the basic anatomical definition of the MDB, detailing its location between suboccipital muscles and the spinal dura mater, its fiber origins, and its role in forming the MDBC. This provides a foundational understanding of the structure. Next, we presented the comparative anatomical evidence of the MDB's conservation across species, followed by its established physiological functions, highlighting its importance in various biological processes. Finally, we delved into the MDB's composition, specifically its collagen - based structure for force transmission, and then introduced its developmental process in rat embryos and human fetuses. This sequence ends with the speculation on the role of mechanical stress in MDB maturation and the need for further research, creating a clear flow from general anatomy and function to specific structural and developmental details. We believe this restructuring effectively improves the logical clarity of the text.

MM�1. Power Analysis for Sample Size Justification

In the "Materials and Methods(Sample Size Determination)" section, we have included a power analysis to justify the sample sizes used. We employed G*Power 3.1.9.7 for this purpose. Under the "t tests" family, we selected the "Correlation: Point biserial model" and adopted an a priori approach.

We set the significance level (α) at 0.05, aiming for a 5% probability of a Type I error. The desired statistical power (1 - β) was set at 0.8, meaning we strived for an 80% chance of detecting a true effect. Based on relevant prior studies, we estimated the effect size to be 0.5. Given our lack of directional assumptions, a two - tailed test was chosen.

Upon inputting these parameters into the software and executing the calculation, the output revealed that a total sample size of 26 was required. The non - centrality parameter (δ) was 2.9439203, the critical t - value was 2.0638986, and the degrees of freedom (Df) were 24. The actual power calculated by the software was 0.8063175, which was close to our target power of 0.8. Considering potential animals dropout, we planned to use 32 mice per group to ensure sufficient statistical power. This addition provides a more robust scientific basis for our sample size determination.

2.Addition of an Untreated Control Group

To rule out non - specific effects from the injection procedure, we added a sham group. This group underwent the same procedures as the shRNA-ITGA7 group, except that it only received needle insertion without any injection. We conducted a morphological analysis on this sham group and compared the results with the shRNA - NC group. As expected, no significant differences were observed between the sham group and the shRNA-NC group. Given that the primary focus of our study is to investigate the specific role of ITGA7 in MDB formation through lentivirus-mediated gene knockdown, the subsequent experiments were mainly designed to explore the differences between the shRNA - NC group and shRNA - ITGA7 group, so we did not include this group in the subsequent experiments.

3. Quantitative Analyses of Picrosirius Red and Masson Staining Images

Regarding the Picrosirius Red and Masson staining images, we have carried out quantitative analyses of collagen fiber density or optical density measurements. We used image J to analyze the images. In the "Materials and Methods" section, we have described the specific steps of the image analysis. The results of these quantitative analyses are presented in the "Results" section, which will enable a more objective and accurate assessment of the changes in collagen fiber content or distribution among different groups.�S4C and S5E Figs)

R Conduct a functional enrichment analysis using Gene Ontology (GO) or KEGG pathway to identify other relevant genes. This will strengthen the conclusions about molecular players in MDB development.

We sincerely appreciate your insightful suggestion regarding the functional enrichment analysis. Given that we had to remove the transcriptomic data section from the current manuscript due to its limited relevance to the main research focus (as previously explained). This is exactly in line with our future research plan. As we mentioned in the limitations section, we are committed to conducting a detailed analysis of the MDB development - related transcriptomic data. In the follow-up studies, we will definitely perform GO and KEGG pathway enrichment analyses to systematically identify other genes associated with MDB development, which will strengthen our understanding of the molecular mechanisms involved. This comprehensive analysis will be a crucial part of our future research.

Discussion:

1.Discuss the relevance of these additional genes potential limitations, such as extrapolating findings from rats to humans and possible interactions with other extracellular matrix proteins that might influence MDB development.

We have added a dedicated paragraph on limitations in the Discussion section, which covers: (1) considerations for translating findings from rodent to human anatomical structures; (2) potential interactions with other extracellular matrix components; (3) the need for a comprehensive exploration of transcriptomic data to fully elucidate the molecular mechanisms of MDB development. These additions provide a more comprehensive and objective perspective while maintaining the focus on our core finding regarding the role of ITGA7 in the development of the myodural bridge (MDB).

2.Provide a more detailed discussion on the potential clinical implications of the findings, including the role of the MDB in CSF circulation and its relationship to cervicogenic headaches.

In the discussion section, we have supplemented a more detailed exploration of the potential clinical implications of the findings, particularly focusing on the role of the myodural bridge (MDB) in cerebrospinal fluid (CSF) circulation and its association with cervicogenic headaches. This study mainly aims to clarify the structural effects of ITGA7 knockdown on MDBC development. It is worth noting that in our ongoing follow-up studies, we will systematically explore the correlations between MDBC developmental abnormalities, CSF circulation, pain mechanisms, and their clinical significance. These studies will include behavioral pain assessments and evaluations of relevant indicators in animal models to further validate the clinical significance of these findings.

Reviewer #2:

1.It is not sufficiently clear how the sample size for each group was calculated.

We greatly appreciate your valuable suggestions. All experiments were performed at least in triplicate, and some were conducted with four times to clarify the results more effectively. For Masson staining and Sirius red staining, we used different sections from the same animals. The mRNA expression of ITGA7 at different developmental stages (E16, P0, P7, and P14) was analyzed in wild-type Sprague-Dawley (SD) rats, qRT-PCR were performed in triplicate, so the number of animals is 12. The Sham group included 4 animals to exclude potential confounding effects caused by needle insertion.

We have rechecked the number of cases for each experiment, and the specific quantities are shown in the following table:

Experimental Method Number of Animals in shRNA - ITGA7 Group Number of Animals in shRNA - NC Group Notes

Initial Enrollment 38 32 Pre-GFP screening

Valid GFP+ Samples (Final) 35 29 RCDmi coverage

qRT - PCR 3 3

Western blotting shRNA - 1

shRNA - 2

shRNA-3, each 3 3 Three samples for each of shRNA - 1 and shRNA - 2 were not included in the total sample size.

Histological sectioning and staining shRNA - 1: 1

shRNA - 2: 1

shRNA-3: 1 P7: 1 In morphology, shRNA-1 and shRNA-2 used for screening effective targets were not included in the total sample size.

Histological sectioning and staining P7: 3

P14: 4 P7: 3

P14: 4 Masson and Sirius red staining

Immunostaining 3 3 Independent cohort

Uniaxial tensile test 3 3 Independent cohort

Scanning electron microscopy 3×2�two time points� 3×2�two time points

Transmission microscopy 3 3 Independent cohort

Total 35 29

In the "Materials and Methods(Sample Size Determination)" section, we have included a power analysis to justify the sample sizes used. We employed G*Power 3.1.9.7 for this purpose. Under the "t tests" family, we selected the "Correlation: Point biserial model" and adopted an a priori approach.

We set the significance level (α) at 0.05, aiming for a 5% probability of a Type I error. The desired statistical power (1 - β) was set at 0.8, meaning we strived for an 80% chance of detecting a true effect. Based on relevant prior studies, we estimated the effect size to be 0.5. Given our lack of directional assumptions, a two - tailed test was chosen.

Upon inputting these parameters into the software and executing the calculation, the output revealed that a total sample size of 26 was required. The non - centrality parameter (δ) was 2.9439203, the critical t - value was 2.0638986, and the degrees of freedom (Df) were 24. The actual power calculated by the software was 0.8063175, which was close to our target power of 0.8. Considering potential mouse dropout, we planned to use 32 mice per group to ensure sufficient statistical power. This addition provides a more robust scientific basis for our sample size determination.

In order to meet the requirements of the ARRIVE guidelines, the Abstract summarizes the research objectives, animal species, strains, and genders, key methods, main findings, and research conclusions. The Introduction includes the scientific background and explanations of the animal species and models used. The research questions and objectives are clearly described. Regarding the ethical statement, our study has obtained approval from the Animal Ethics Review Committee of Dalian Medical University (No: AEE21085). Interventions to reduce pain are included in the experimental procedures. The results are interpreted in relation to the research objectives and hypotheses, and the limitations are clarified. The manuscript contains data access statements and a declaration of interests.

ARRIVE guidelines

Study design

1 For each experiment, provide brief details of study design including:

a.The groups being compared, including control groups. If no control group has been used, the rationale should be stated.

b. The experimental unit (e.g. a single animal, litter, or cage of animals) Materials and methods

Animal model and Housing

(Lines 141-155)

Sample size

2 a. Specify the exact number of experimental units allocated to each group, and the total number in each experiment. Also indicate the total number of animals used. b. Explain how the sample size was decided. Provide details of any a priori sample size calculation, if done. Materials and me

---

## [Decision Letter · Decision Letter 1]

12 Jul 2025

Dear Dr. Ma,

We look forward to receiving your revised manuscript.

Kind regards,

Aldo Corriero, Ph.D.

Academic Editor

PLOS ONE

Journal Requirements:

Reviewers' comments:

Reviewer's Responses to Questions

**Comments to the Author**

Reviewer #1: All comments have been addressed

Reviewer #2: All comments have been addressed

Reviewer #3: All comments have been addressed

2. Is the manuscript technically sound, and do the data support the conclusions?

Reviewer #1: Yes

Reviewer #2: Yes

Reviewer #3: Yes

3. Has the statistical analysis been performed appropriately and rigorously?

Reviewer #1: Yes

Reviewer #2: Yes

Reviewer #3: Yes

4. Have the authors made all data underlying the findings in their manuscript fully available?

Reviewer #1: Yes

Reviewer #2: Yes

Reviewer #3: Yes

5. Is the manuscript presented in an intelligible fashion and written in standard English?

Reviewer #1: No

Reviewer #2: Yes

Reviewer #3: Yes

Reviewer #1: The authors are to be congratulated for conducting a comprehensive and methodologically rigorous study addressing an important anatomical and biomechanical question. The manuscript presents convincing data demonstrating that ITGA7 is essential for the force-dependent development and maturation of the myodural bridge in neonatal rats.

Points to address before final acceptance:

1. Correct minor typographical errors, such as "invlove" → "involve."

2. Introduction: Residual focus on CSF dynamics, though reduced, could distract from the central mechanobiology hypothesis. Ensure all speculative statements regarding CSF dynamics and clinical analogies are clearly presented as hypotheses for future research rather than conclusions.

3. Consider adding in Results a summary table consolidating main quantitative results (e.g., collagen volume fraction, muscle force, gene expression) to aid reader comprehension.

4. Discusion: Discussion on CSF dynamics and clinical analogies (e.g., Chiari malformations) should be presented as speculative, not definitive conclusions. Potential sex-based differences (given mixed-sex animals) are not addressed.

Reviewer #2: In my opinion, the manuscript is ready for publication in PlosONE. The decision to remove the transcriptomic data, although significant, is well justified and does not affect the main conclusions of the study. The authors have demonstrated a serious commitment to improving the manuscript's quality and have adequately addressed all reviewers' concerns.

Reviewer #3: The authors have satisfactorily answered the questions raised by this reviewer. The manuscript has been extensively revised, incorporating the suggestions of the reviewers and modifying some of the sentences in such a way that the reading is more agile. The technical quality of some of the techniques used must be highlighted, as well as the quality of the images that illustrate the results. It is also important to note that myodural bridges, at least in the human species, have proprioceptive innervation and that sensory nerve formations have been described in them. I leave it to the discretion of the authors to include this information in the final version of the manuscript.

**Do you want your identity to be public for this peer review?** For information about this choice, including consent withdrawal, please see our Privacy Policy

Reviewer #1: No

Reviewer #2: No

Reviewer #3: No

---

## [Author Response · Author response to Decision Letter 2]

16 Jul 2025

Reviewer #1:

We sincerely appreciate your constructive comments, which have greatly enhanced the quality of our manuscript. All suggestions have been carefully addressed as follows:

1.Correct minor typographical errors, such as "invlove" → "involve."

All minor typographical errors, including "invlove" corrected to "involve," have been identified and revised in the manuscript.

2.Introduction: Residual focus on CSF dynamics, though reduced, could distract from the central mechanobiology hypothesis. Ensure all speculative statements regarding CSF dynamics and clinical analogies are clearly presented as hypotheses for future research rather than conclusions.

In the Introduction, we have rephrased all statements related to CSF dynamics and clinical analogies to explicitly frame them as hypotheses for future research, rather than conclusions, to avoid distracting from the central mechanobiology focus.

①“Notably, the upper cervical region harharbors a specialized biomechanical microenvironment critical for localized CSF regulation— the myodural bridge (MDB)[8-11], et al.”→“Notably, the upper cervical region harharbors a specialized biomechanical microenvironment that may be involved in localized CSF regulation— the myodural bridge (MDB)[8-11], ”

3.Consider adding in Results a summary table consolidating main quantitative results (e.g., collagen volume fraction, muscle force, gene expression) to aid reader comprehension.

A summary table has been added in the Results section.(S3 Table)

4.Discussion: Discussion on CSF dynamics and clinical analogies (e.g., Chiari malformations) should be presented as speculative, not definitive conclusions. Potential sex-based differences (given mixed-sex animals) are not addressed.

In the Discussion, we have adjusted the content on CSF dynamics and clinical analogies (e.g., Chiari malformations) to emphasize their speculative nature. Additionally, In the limitations of the discussion, we clarify that our mixed-sex animal model cannot analyze potential sex-based differences in the role of ITGA7 in MDB development.

①"This study provides further evidence that suboccipital muscle force is transmitted to the SDM via the MDB, acting as a dynamic driver of CSF circulation."→"This study suggests that suboccipital muscle force transmitted to the SDM via the MDB may act as a potential dynamic driver of CSF circulation."

②"The potential impact of this condition may be similar to that of cerebellar tonsillar herniation, known to disrupt CSF circulation."→"It is plausible that this condition could have effects analogous to cerebellar tonsillar herniation (a disorder associated with CSF circulation disruption), though such a link remains speculative and requires direct investigation."

③"Numerous studies have demonstrated that the MDB couples suboccipital muscle movements to cerebrospinal fluid (CSF) dynamics in human[11, 44] and animal models..."→"Numerous studies suggest that the MDB may couple suboccipital muscle movements to cerebrospinal fluid (CSF) dynamics in humans[11, 44] and animal models...".

④"These findings provide direct evidence for the role of the MDBC in headache... This preserves mechanical buffering, maintains CSF circulation, and reduces risks of headache and CSF leakage from CSF dynamic disorders[60]."→"These findings suggest a potential role of the MDBC in headache... This may help preserve mechanical buffering, support CSF circulation, and reduce potential risks of headache and CSF leakage related to CSF dynamic disorders[60], though clinical confirmation is needed."

Reviewer #2:

Thank you very much for your positive comments and valuable feedback on our manuscript. We truly appreciate your time and efforts in reviewing our work.

Reviewer #3:

We sincerely appreciate your valuable insight regarding the proprioceptive innervation of myodural bridges (MDBs). This is indeed an important aspect, and thank you for highlighting it. Our research team is currently conducting studies related to this area, and relevant articles will be published subsequently. We truly value your expertise and hope to further explore this interesting topic in future research.

Thank you again for your valuable contributions. We hope these revisions address your concerns adequately. Please let us know if further adjustments are needed.

Best regards,

Wei Ma

Hong-jin Sui

---

## [Editor Report · Decision Letter 2]

22 Jul 2025

Force-dependent development of the myodural bridge in rats: the impact of Integrin α7

PONE-D-25-11068R2

Dear Dr. Ma,

We’re pleased to inform you that your manuscript has been judged scientifically suitable for publication and will be formally accepted for publication once it meets all outstanding technical requirements.

Kind regards,

Aldo Corriero, Ph.D.

Academic Editor

PLOS ONE

Additional Editor Comments (optional):

All the comments have been properly addressed and the manuscript can be accepted for publication
---

## [Editor Report · Acceptance letter]

PONE-D-25-11068R2

PLOS ONE

Dear Dr. Ma,

I'm pleased to inform you that your manuscript has been deemed suitable for publication in PLOS ONE. Congratulations! Your manuscript is now being handed over to our production team.

Kind regards,

on behalf of

Dr. Aldo Corriero

Academic Editor

PLOS ONE